# Impaired LEF1 Activation Accelerates iPSC-Derived Keratinocytes Differentiation in Hutchinson-Gilford Progeria Syndrome

**DOI:** 10.3390/ijms23105499

**Published:** 2022-05-14

**Authors:** Xiaojing Mao, Zheng-Mei Xiong, Huijing Xue, Markus A. Brown, Yantenew G. Gete, Reynold Yu, Linlin Sun, Kan Cao

**Affiliations:** 1Department of Cell Biology and Molecular Genetics, University of Maryland, College Park, MD 20817, USA; xmao1123@terpmail.umd.edu (X.M.); zhengmei.xiong@nih.gov (Z.-M.X.); hjxue05@umd.edu (H.X.); mbrown97@mgh.harvard.edu (M.A.B.); ygete@umd.edu (Y.G.G.); reyy@terpmail.umd.edu (R.Y.); linlinsun2020@zju.edu.cn (L.S.); 2National Human Genome Research Institute, National Institutes of Health, Bethesda, MD 20892, USA; 3Key Laboratory of Diagnosis and Treatment of Aging and Physical-Chemical Injury Diseases of Zhejiang Province, The First Affiliated Hospital, School of Medicine, Zhejiang University, Hangzhou 310027, China

**Keywords:** HGPS, progeria, lamin A, WNT signaling, LEF1, skins, epidermal development, keratins

## Abstract

Hutchinson–Gilford progeria syndrome (HGPS) is a detrimental premature aging disease caused by a point mutation in the human *LMNA* gene. This mutation results in the abnormal accumulation of a truncated pre-lamin A protein called progerin. Among the drastically accelerated signs of aging in HGPS patients, severe skin phenotypes such as alopecia and sclerotic skins always develop with the disease progression. Here, we studied the HGPS molecular mechanisms focusing on early skin development by differentiating patient-derived induced pluripotent stem cells (iPSCs) to a keratinocyte lineage. Interestingly, HGPS iPSCs showed an accelerated commitment to the keratinocyte lineage than the normal control. To study potential signaling pathways that accelerated skin development in HGPS, we investigated the WNT pathway components during HGPS iPSCs-keratinocytes induction. Surprisingly, despite the unaffected β-catenin activity, the expression of a critical WNT transcription factor LEF1 was diminished from an early stage in HGPS iPSCs-keratinocytes differentiation. A chromatin immunoprecipitation (ChIP) experiment further revealed strong bindings of LEF1 to the early-stage epithelial developmental markers K8 and K18 and that the LEF1 silencing by siRNA down-regulates the K8/K18 transcription. During the iPSCs-keratinocytes differentiation, correction of HGPS mutation by Adenine base editing (ABE), while in a partial level, rescued the phenotypes for accelerated keratinocyte lineage-commitment. ABE also reduced the cell death in HGPS iPSCs-derived keratinocytes. These findings brought new insight into the molecular basis and therapeutic application for the skin abnormalities in HGPS.

## 1. Introduction

Hutchinson–Gilford Progeria syndrome (or HGPS, progeria) is an ultra-rare but devastating genetic disorder that causes premature aging in children [1]. The HGPS patients appear normal at birth but soon develop a series of severe health conditions that occur in the process of aging, including alopecia, scleroderma, subcutaneous fat loss, bone, and joint defects, and cardiovascular diseases, etc. Most HGPS patients die of arteriosclerosis at an average age of 14.5 [2,3]. Classical HGPS is caused by a de novo point mutation within the human *LMNA* gene (c.1824C > T). This gain-of-function mutation activates a cryptic splice site that generates a truncated, unproperly processed lamin A precursor that retains a hydrophobic farnesyl tail named progerin [1,4]. The accumulation of progerin in the cell nucleus induces abnormal nuclear morphology and altered gene expression and chromatin structure [5,6,7]. However, how progerin accelerates the aging process in many tissues and organs during their development remains largely unclear, particularly in the skin, the human body’s largest organ.

Human skin consists of the epidermis, dermis, and subcutaneous tissues [8]. In HGPS, skin abnormalities, including reduced epidermis thickness, alopecia, dimpling, wrinkling, altered pigmentation, and decreased subcutaneous fat, overlap significantly with signs of normal aging [2,3]. As the outermost surface of human skin, the epidermis is susceptible to environmental stimuli and develops symptoms of aging. Keratinocytes are the primary cell type in the stratified epidermis. Moving outward from the basement membrane, the basal layer (*Stratum basale*) keratinocytes undergo a sequential differentiation process to form the spinous layer (*Stratum spinosum*), granular layer (*Stratum granulosum*), and cornified layer (*Stratum corneum*) above [9].

HGPS skin abnormalities have been studied in transgenic mice with inducible epidermal expression of progerin under the control of Keratin 5 (K5) or Keratin 14 (K14) promotors [10,11]. Postnatal expression of progerin under K5 promoter resulted in premature exhaustion of adult stem cells in mouse skins [12]. Epidermal keratinocytes with aberrant proliferation capacity and altered gene expression have also been found in these animals [10,12]. In the same animal model with embryonic induction of progerin expression, keratinocytes from the interfollicular epidermis (IFE) presented an abnormal tendency to divide symmetrically, generating excessive basal keratinocytes at the basement membrane. As a result, basal cells were often spotted in suprabasal layers of the epidermis in HGPS transgenic mice and were likely to induce epidermal hyperplasia [10,13,14]. Pathway analysis indicated that impaired WNT/β-catenin signaling may be responsible for this phenomenon, as progerin disrupted the nuclear localization of emerin and nesperin-2, impeding the nuclear transportation of β-catenin [13,15]. WNT/β-catenin pathway inhibition was also found in human and mouse progeroid cells, which leads to a reduction in extracellular matrix synthesis that results in cell proliferation arrest and apoptosis [16]. However, in another study using mice expressing progerin under the K14 promoter, the skin phenotype was not developed despite the nuclear abnormalities in keratinocytes [11]. While these transgenic animals with lineage-specific progerin expression served as great models to study skin abnormalities in HGPS, they may not fully represent the symptoms in HGPS patients due to genetic divergence between humans and mice. Moreover, these studies focused on keratinocytes from fully developed mouse skins, and thus only those phenotypes at late developmental stages were characterized. In order to investigate the HGPS disease progression in early epidermal development with a humanized model, we took advantage of the patient-derived iPSCs and monitored the full differentiation process towards a keratinocyte lineage.

Currently, there is no cure for HGPS. Yet, the recent application of Adenine Base Editing (ABE) in HGPS primary cells and an animal model offered up-and-coming results [17]. This novel gene-editing tool linked a deoxyadenosine deaminase to a deactivated CRISPR vector, which conducts A•T to G•C base pairs conversion at the targeted genome region [18]. In HGPS, with the precise-targeting single-guide RNA (sgRNA), these base editors directly corrected the *LMNA* c.1824C > T mutation with high efficiency. In vitro base editing in HGPS primary fibroblasts and endothelial cells showed complete progerin clearance, while the postnatal ABE-AAV9 injection greatly improved the health of HGPS mice and significantly extended their lifespans [17,19].

In this study, we investigated the early development of keratinocytes in HGPS. We first differentiated a pair of well-characterized HGPS patients and the healthy, unaffected father (normal)-derived induced pluripotent stem cells (iPSCs) into a keratinocyte lineage [19,20,21,22,23]. We found that during the differentiation process, HGPS iPSCs showed an accelerated commitment to the keratinocyte lineage. Furthermore, we identified a possible upstream effector in epidermal development: LEF1, whose activation was diminished in HGPS iPSCs differentiation. By utilizing the novel gene-editing technique ABE, we partially corrected the genetic mutation in HGPS during the keratinocytes induction and rescued the defects in HGPS iPSCs-derived keratinocytes.

## 2. Results

### 2.1. Differentiating Normal and HGPS-Derived iPSCs into Keratinocytes

To differentiate normal and HGPS patient-derived iPSCs into the keratinocyte lineage in vitro, we adapted a four-week differentiation protocol including two inductions with retinoid acid (RA) and BMP-4 at the initial stage, followed by keratinocyte growth medium changes every other day for up to 28 days [24,25]. At the end of differentiation, a subpopulation of basal keratinocyte-like cells with cobblestone morphology was observed (Figure 1A). To evaluate the efficacy of differentiation, we first detected the expression of lamin A/C, progerin, and basal keratinocyte markers ∆Np63 and K14 by immunofluorescence staining (Figure 1B,C). As expected, all these iPSCs-derived cells showed ∆Np63 and K14 expression with different fluorescence intensities. Their cellular distribution was also comparable to those in primary keratinocytes (Figure 1C). Progerin expression was only detected in cells differentiated from HGPS iPSCs (Figure 1B).

Conventionally, a succession of keratin genes expressed in keratinocytes has been identified during mammalian epidermal development: in the early embryonic stage, the expression of keratin pair K8–K18 peaks in single-layered epithelial progenitors that give rise to the epidermis and declines rapidly towards the keratinocyte commitment [26]. Afterward, with the induction of p63, these progenitor cells are committed to the epidermal fate by turning on K5–K14 expression [27,28]. These p63+, K5/K14+ cells are adult stem cells that can proliferate as well as differentiate into spinous cells, granular cells, and squames by upward stratification [29]. One keratin type that marks this terminal differentiation is K1 [30,31]. To characterize the normal and HGPS iPSCs-keratinocytes differentiation at each differentiation stage, the expression of these markers was quantified weekly for four weeks (Figure 2A,B and Appendix A) with quantitative RT-PCR and Western blotting analysis. We found that Lamin A expression was gradually upregulated during the four-week differentiation and only detected progerin expression in HGPS samples (Figure 2A and Appendix A). The early marker K8 was first detected upon the keratinocytes induction, and, interestingly, its expression declined more rapidly in HGPS iPSCs-keratinocytes differentiation compared to the normal situation. Accordingly, basal layer keratinocyte markers ∆Np63 and K14 expression increased through normal and HGPS iPSCs differentiation. At each time point detected, ∆Np63 and K14 levels were consistently higher in HGPS than the normal control. The terminal differentiation marker K1 was only seen in HGPS iPSCs-derived keratinocytes at week 4 but absent in normal iPSCs differentiation, suggesting that only the 4-week differentiated population in HGPS contained the fully matured keratinocytes to initiate the terminal differentiation (Figure 2A,B and Appendix A).

### 2.2. Dysregulated Cell Cycle in HGPS iPSCs-Keratinocytes Differentiation

Progerin disrupts the cell cycle progression in mammalian cells [23,32,33]. To investigate the cell cycle partitioning in HGPS epidermal development, we first labeled the normal and HGPS cells at the intermediate (week 2) and late (week 4) iPSCs-keratinocytes differentiation stages using Bromodeoxyuridine (BrdU) as an indication for the DNA synthesis at the S phase. These cells were then stained with fluorescent BrdU antibody and propidium iodide (PI) and quantified for frequencies at each cell cycle stage by a flow cytometry analysis (Appendix A and Figure 3A). Notably, we observed a more significant fraction of HGPS cells at the G2/M phase than normal differentiation at the intermediate stage. This disparity became more significant at the late stage of differentiation (Figure 3A).

Because the cell cycle abnormalities in differentiating HGPS epidermal cells resembles our previous findings in HGPS iPSCs-derived smooth muscle cells (iSMCs), which is likely linked with mitotic catastrophe induced by progerin [23,34], we next asked if there were increases in cell death during the HGPS iPSCs-keratinocytes induction. To address this question, we performed an Annexin V–PI assay using cells at the intermediate and late stages of differentiation. The population sorted as Annexin V-positive and PI-negative (early apoptotic cells) was quantified as an indicator for cell death (Figure 3B). We observed that cell death in both normal and HGPS differentiation increased from the intermediate stage (week 2) to the late stage (week 4). No significant difference in the apoptotic rate and senescence marker p16 expression was detected in HGPS cells (Figure 3C and Appendix A).

### 2.3. LEF1 Is Down-Regulated in Early HGPS iPSCs-Keratinocytes Differentiation

Previous studies identified WNT signaling as one of the most affected pathways in mouse skin keratinocytes with epidermal-specific progerin expression [13]. WNT signaling is an early morphogenetic pathway that has been activated upon gastrulation and orchestrates multiple processes in the skin development [29,35]. An impaired canonical WNT pathway with diminished nuclear β-catenin localization has been reported in progeroid mouse skins and HGPS osteoblast differentiation [13,15]. Given that, we aimed to examine if the accelerated keratinocyte lineage-commitment in HGPS iPSCs differentiation comes along with an altered WNT pathway.

We first checked the β-catenin level by Western blot analysis in iPSCs-keratinocytes differentiation. We detected the weekly expression of the total and the active form of β-catenin throughout the differentiation process (Figure 4A). Our results indicated that β-catenin expression was activated from a very early stage upon the iPSCs-keratinocytes induction and was decreased over time under both normal and HGPS conditions. At each week, the β-catenin level in HGPS was quite comparable to normal conditions (Figure 4A and Appendix A). Because progerin was reported to disrupt the nuclear translocation of β-catenin [13,15], we then performed nuclear extraction for the early-stage (week 1) differentiating cells with high β-catenin expression. Still, no significant difference in active β-catenin expression was detected in the HGPS nucleus (Figure 4B and Appendix A). We then asked if the transcription of those essential WNT pathway genes was activated upon the early iPSCs-keratinocytes induction. Indeed, despite the unchanged TCF7 mRNA level, significant upregulations of LEF1, TCF7L1, and TCF7L2 transcription were detected at differentiation day 5, following the second stimuli with BMP-4 and RA. Interestingly, the LEF1 mRNA level in HGPS did not catch up with normal conditions at differentiation day 5, while no other gene expression showed the same trend (Figure 4C). We further detected the weekly expression of LEF1 throughout the differentiation process. Again, diminished LEF1 expression was observed in HGPS cells at all differentiation stages (Figure 4D,E).

### 2.4. LEF1 Regulates Keratinocytes Differentiation through K8/K18

To investigate if LEF1 is an upstream regulator that determines the iPSCs’ fate towards keratinocyte lineage upon RA and BMP-4 stimuli, we performed a genome-wide search for potential LEF1 binding targets among genes highly expressed in early epidermal development. At the gene locus of epidermal progenitor markers K8 and K18, which are adjacent to each other at chromosome 12, we identified seven putative LEF1 binding sites (Figure 5A). We then performed a Chromatin immunoprecipitation (ChIP) assay using LEF1 antibody and quantified the percent input of these binding sites by qPCR. Our results indicated strong LEF1 recruitment at binding sites K8-1, K8-5, and K8-6, which correlates with LEF1 ChIP-seq peaks in HEK293T and human embryonic stem cells (HUES64)-derived mesoderm (Figure 5B and Appendix A). LEF1 binding at K8-1 was the strongest among them compared to a positive control at the *MYC* gene (Figure 5B). Due to the decreased expression of LEF1 in HGPS iPSCs-keratinocytes, we were unable to use the LEF1 antibody to IP a sufficient amount of genomic DNA for quantitative PCR analysis.

To further test the impact of LEF1 on K8/K18 expression, we down-regulated LEF1 expression at differentiation day 5 using a combination of three LEF1-targeting siRNAs. Our results showed that with a partial knockdown of LEF1 (Figure 5C,D), both K8 and K18 mRNA transcription decreased significantly (Figure 5E). Changes in their protein expression were not detected at this time point based on the Western blot analysis (Appendix A).

### 2.5. Correcting the HGPS Mutation in iPSCs-Keratinocytes Differentiation with ABE

To revert the *LMNA* c.1824 C > T mutation and curb its impact on HGPS iPSCs-derived keratinocytes from an early developmental stage, we conducted Adenine base editing (ABE) using the c.1824 C > T targeting ABE7.10max lentiviral vector at differentiation day 5, followed by three days of puromycin selection. ABE7.10max vector with human non-targeting sgRNA was included (mock-corrected) (Figure 6A). At the end of iPSCs-keratinocytes differentiation (week 4), about 50% of *LMNA* c.1824 T (pathogenic) was corrected to C (wild-type) (Figure 6B). Although progerin was still detectable in HGPS-corrected iPSCs-keratinocytes, immuno-stained cells with very bright nuclei consisting of highly expressed progerin were rarely seen after ABE correction (Figure 6C). Our RT-qPCR results showed a 50% reduction in progerin mRNA transcription in HGPS-corrected iPSCs-derived keratinocytes (Figure 6D) and that the ABE treatment led to a significant decrease in progerin expression (Figure 6E). Interestingly, the keratinocyte markers expression in HGPS-corrected cells was significantly reversed despite the moderate correction efficiency. Down-regulated p63, K14, and K1 and up-regulated K8 and K18 expression were detected after the base editing (Figure 6F and Appendix A). At differentiation week 4, the mRNA transcription of LEF1 was slightly restored in HGPS-corrected iPSCs-keratinocytes (Figure 6F). Moreover, the phenotype of massive cell death was alleviated in HGPS-corrected iPSCs-derived keratinocytes (Figure 6G), although the cell cycle partitioning remained unchanged (Appendix A).

## 3. Discussion

### 3.1. Stem Cell Depletion May Contribute to the Accelerated Keratinocyte Commitment in HGPS iPSCs Differentiation

In 2007, Halascheck–Wiener et al. promoted a stem cell exhaustion theory to explain the segmental aging phenotypes that appears to be more severe in tissues under continuous strong mechanical stress or endure high turnover rate in HGPS [36]. For instance, the nuclear accumulation of progerin in HGPS endothelial cells and smooth muscle cells induces genome instability and increases cell death and turnover rate, which may result in a depletion of endothelial progenitor cells (EPCs) [23,36,37]. The continuing loss of EPCs will impair vascular maintenance and regeneration, causing severe cardiovascular conditions [38,39]. The stem cell exhaustion theory may also explain the hair loss and lipodystrophy in HGPS [36]. Moreover, epidermal hyperplasia and premature depletion of adult stem cells were reported in mouse skins with tissue-specific progerin expression [12]. In this study, we observed an accelerated commitment of HGPS iPSCs to keratinocyte lineage, based on the keratinocyte development markers at early, middle, and late stages. Consistent with studies in other HGPS cell types, these differentiating HGPS keratinocytes showed a higher percentage of G2/M cells. While we did not detect a statistically significant increase in cell death in HGPS in comparison to the normal control when we analyzed three biological replicates together, we consistently noted that in each of our replications, HGPS samples showed a higher death rate than control cells. The lack of statistical significance is at least partially due to the experimental variations during each differentiation experiment. Together, these results suggest that HGPS keratinocytes differentiation is accelerated, resulting in undesired, hastened cell turnover and eventual depletion of epithelial stem cells.

### 3.2. LEF1 Is an Early Regulator in Epidermal Development, and Its Down-Regulation Accelerates the Keratinocyte Lineage Commitment in HGPS

WNT signaling is one of the most affected pathways in the HGPS mouse keratinocytes [13]. In our iPSCs-keratinocytes differentiation experiments, many WNT pathway components were highly expressed upon the induction of RA and BMP-4, including LEF1, TCF7L1, and TCF7L2, etc. Interestingly, we found that the expression of a critical WNT transcription factor LEF1 was diminished at an early stage in HGPS iPSCs differentiation. Moreover, the HGPS iPSCs-derived cells maintained a lower LEF1 expression level throughout the epidermal development than the normal control. The inhibited canonical Wnt pathway due to reduced LEF1 nuclear localization and transcriptional activity was also found in progeroid cells in both humans and mice [16]. We further investigated the impact of LEF1 on the expression of epidermal progenitor genes K8 and K18. In iPSCs-keratinocytes differentiation, K8/K18 expression was induced upon RA treatment and decreased over time significantly after p63 was activated [30]. In HGPS iPSCs-keratinocytes induction, the expression of K8/K18 decreased more rapidly, possibly due to an insufficient LEF1 level, thus contributing to an accelerated commitment towards the p63 and K5/K14-positive keratinocyte lineage (Figure 7B). Overexpressing LEF1 in HGPS iPSCs-keratinocytes will be a complementary approach to further assess its impact on K8/K18 expression and keratinocyte lineage commitment in the future.

Previous studies revealed a β-catenin loss in the HGPS cell nucleus due to disrupted nuclear transportation caused by progerin [13,15]. In our iPSCs-keratinocytes differentiation process, the nuclear β-catenin level was unchanged at the early stage of iPSCs-keratinocytes differentiation, when progerin aggregates were not yet formed at the nuclear envelope. Although it is unclear how LEF1 transcription was inhibited at this stage of differentiation, other potential pathways regulate LEF1 expression in a β-catenin-independent manner, including the BMP pathway, which activates LEF1 transcription via P-SMAD in the neural stem cells differentiation [40]. These pathways, together with components in WNT pathways, warrant further analysis to decipher the LEF1 transcriptional inhibition in HGPS iPSCs-keratinocytes differentiation.

### 3.3. ABE in HGPS iPSCs-Keratinocytes Differentiation

Since the developmental defects in HGPS iPSCs-keratinocytes have an early onset, we strived to correct the c.1824C > T mutation at an early developmental stage without disturbing the initial inductions with RA and BMP-4. Our Adenine base editing in HGPS iPSCs-derived keratinocytes did not reach the high correction efficiency and complete progerin clearance as in endothelial cells. Notably, in HGPS animals with postnatal ABE-AAV9 virus injection, the ABE efficiency in the skin was also lower than in other tissues such as the liver and heart [17]. One possible reason for low ABE efficiency in HGPS iPSCs-keratinocytes lies in the cell-cycle abnormalities in HGPS epidermal development, due to progerin. According to the cell-cycle analysis in normal and HGPS cells, at differentiation week 2, when ABE had its optimal effects, a reduced S-phase population made it difficult for genome editing to occur. Another unique feature in epidermal development may also reduce the base editing efficiency: the high cell death and turn-over rate. Massive cell death was observed in normal iPSCs-keratinocytes differentiation, and ABE only slightly reduced the cell death in HGPS-corrected cells. Given that, cells may still undergo a high level of cell death even with a corrected genotype.

In addition, because progerin was relatively stable, it may persist in the HGPS-corrected cells long after the mutation has been corrected. Thus, it may take a long time for complete progerin clearance. However, as basal layer keratinocytes have limited self-renewal potential and start spontaneous terminal differentiation under the in vitro culture condition, there was only a short window to evaluate the properties of HGPS-corrected cells. Interestingly, our results indicated that even a tiny portion of correction makes a large difference in decelerating the commitment towards keratinocytes in HGPS. A similar phenomenon was seen in ABE-corrected HGPS animals. Despite the small percentage of correction in many tissues, their health condition improved greatly with a significantly extended lifespan [17]. Only a slightly higher LEF1 expression was detected at the end of differentiation, indicating a less critical role of LEF1 at the late stage of epidermal development. Future experiments such as ABE in HGPS iPSCs may increase the correction efficiency and progerin clearance in HGPS iPSCs-derived keratinocytes.

## 4. Materials and Methods

### 4.1. Cell Culture and Keratinocytes Differentiation from iPSCs

The HGPS patient and the healthy, unaffected father donor (normal)-derived iPSCs used in this study were reprogramed from skin fibroblasts of a male HGPS patient who carried the classic G608G HGPS mutation (HGADFN167) and his normal father (HGADFN168) and were provided by the Progeria Research Foundation Cell and Tissue Bank. In vitro differentiation protocol towards the keratinocyte lineage was adapted from previous publications with minor modifications [24,25]. In brief, feeder-free iPSCs were cultured in defined keratinocytes serum-free medium (DKSFM, Gibco, Waltham, MA, USA) containing 25 ng/mL bone morphogenetic protein-4 (BMP-4, R&D Systems, Minneapolis, MN, USA) and 1 μM all-trans Retinoid Acid (RA, Sigma-Aldrich, St. Louis, MO, USA) at the initial stage of differentiation (medium was changed once at differentiation day 3). Starting from day 5, DKSFM was changed every other day for up to 28 days. Cells were not passaged during the 28-day differentiation. Fully differentiated keratinocytes can be passed three to five times on Collagen I (Sigma-Aldrich, St. Louis, MO, USA)-coated tissue culture dishes. Normal human primary keratinocytes isolated from neonatal foreskin (HEKn) were purchased from ATCC (PCS-200-010). Cells were maintained at 37 °C and 5% CO_2_ in a humidified incubator.

### 4.2. Adenine Base Editor Lentiviral Vectors Production

The ABE7.10max-VRQR lentiviral vector targeting the HGPS G608G mutation was generated as described in Koblan et al., 2021 [17]. Human non-targeting control vectors were generated as described in Gete et al., 2021 [19].

### 4.3. Immunocytochemistry

For immunocytochemistry, cells were washed twice with PBS and fixed in 4% paraformaldehyde (PFA) for 15 min, followed by permeabilization with 0.5% Triton in PBS for 5 min. Cells were then blocked with 4% BSA in PBS for one hour. After that, cells were incubated with primary antibodies in 4% BSA in PBS overnight at 4 °C. The next day, after five PBS washes, cells were incubated in secondary antibodies diluted in 4% BSA in PBS for one hour. Cells were then washed five times with PBS, and fluorescence images were acquired with a Zeiss LSM 710 confocal microscope (Zeiss International, Oberkochen, Germany). Fluorescence intensity was adjusted with ImageJ software (NIH). The primary antibodies used for immunocytochemistry were: lamin A/C (1:250, Millipore MAB3211), progerin (1:250, Cao et al., 2011 [41]), ∆Np63 (1:250, Biolegend, San Diego, CA, USA #619001), and K14 (1:500, Invitrogen, Waltham, MA, USA MA5-11599). The secondary antibodies used were: Alexa Fluor 488 donkey anti-rabbit IgG (1:1000, Invitrogen), Alexa Fluor 594 donkey anti-rabbit IgG (1:1000, Invitrogen), Alexa Fluor 488 donkey anti-mouse IgG (1:1000, Invitrogen), and Alexa Fluor 594 donkey anti-mouse IgG (1:1000, Invitrogen).

### 4.4. Western Blot

Whole-cell lysates for immunoblotting were prepared by dissolving cells in Laemmli Sample Buffer containing 5% 2-mercaptoethanol (Bio-Rad, Hercules, CA, USA). Nuclear and cytosolic fractions of iPSCs-differentiated cells were acquired using NE-PER™ Nuclear and Cytoplasmic Extraction Reagents (Thermo Fisher Scientific, Waltham, MA, USA #78833) as per the manufacturer’s instructions. Protein samples were loaded on 10–12% polyacrylamide gels and transferred onto 0.45 µm pore-size nitrocellulose membranes (Bio-Rad) using the Turboblot (BioRad). Blots were incubated overnight at 4 °C with primary antibodies and then probed with secondary antibodies before ECL development and imaging (Bio-Rad). The primary antibodies used for immunoblotting are as follows: lamin A/C (1:500, Millipore MAB3211), progerin (1:500, Cao et al., 2011), ∆Np63 (1:500, Biolegend, San Diego, CA, USA #619001), K14 (1:500, Invitrogen MA5-11599), K8 (1:500, Biolegend #904804), K18 (1:1000, Cell Signaling #4548), total β-Catenin (1:1000, Cell Signaling #9562), Non-phospho (Active) β-Catenin (1:1000, Cell Signaling #8814), RCC1 (1:1000, Cell Signaling #3589), S6 Ribosomal Protein (1:1000, Cell Signaling #2217), LEF1 (1:250, Santa Cruz sc-374412), and β-actin (1:1000, Sigma-Aldrich, St. Louis, MO, USA A3854).

### 4.5. RNA Isolation and Real-Time Quantitative PCR

Total genomic RNA was extracted with Trizol (Life Technologies, Carlsbad, CA, USA) and purified using the RNeasy Mini kit (Qiagen, Hilden, Germany) as per the manufacturer’s instructions. The RNA yield was determined by the NanoDrop 2000 spectrophotometer (Thermo Fisher Scientific, Waltham, MA, USA). One µg of total RNA was converted to cDNA using the iScript Select cDNA Synthesis kit (Bio-Rad). Quantitative RT-PCR was performed in triplicate using SYBR Green Supermix (Bio-Rad) on the CFX96 Real-Time PCR Detection System (C1000 Thermal Cycler, Bio-Rad). All primers used in this study are listed in Table 1.

### 4.6. Cell Cycle Analysis

Cell cycle analysis was performed at differentiation week 2 and week 4 in normal and HGPS iPSCs-keratinocytes induction. Before harvest, cells were incubated in a 10 uM BrdU (BD #550891) labeling medium for one hour. The collected cells were then washed once and resuspended in 100 μL PBS. The cell suspension was fixed in 1 mL of 70% (*v*/*v*) ethanol for one hour on ice, followed by a PBS wash and denatured in 2N HCl with 0.5% Triton X-100 for 30 min at RT. Cells were washed again with PBS, blocked, and incubated in Alexa Fluor 488 anti-BrdU antibody solution (1:100, Invitrogen #B35130) at 4 °C overnight. On the next day, the cells were pelleted, rinsed with PBS, and resuspended in 50 μg/mL propidium iodide (PI, Invitrogen) with 100 μg/mL DNase-free RNase (Thermo Scientific) in PBS for 30 min at 37 °C. Flow cytometry was performed with FACS CantoII (BD), and the data were analyzed by FlowJo software.

### 4.7. Apoptosis Assay

PI–annexin V apoptosis assay was performed at differentiation week 2 and week 4 in normal and HGPS iPSCs-keratinocytes induction according to the manufacturer’s instruction (BD). In brief, cells were harvested and rinsed with PBS and then resuspended and stained with 100 μL of 1x annexin V binding buffer containing 5 μL of annexin V and 5 μL of PI for 25 min in the dark at room temperature. Stained samples were analyzed by FACS CantoII (BD), and the data were processed by FlowJo software.

### 4.8. Chromatin Immunoprecipitation

LEF1 ChIP was performed in early-differentiating (week 1) epidermal cells derived from normal iPSCs. Detailed experimental procedures were described in the protocol of SimpleChIP^®^ Plus Sonication Chromatin IP Kit (Cell Signaling #56383). Briefly, protein and DNA were cross-linked using 1% formaldehyde for 10 min. Nuclei were isolated and lysed, and chromatin was sheared to an average size of 400 bp by sonication. A small aliquot of the supernatant was used as input control, and the remaining sonicated chromatin was divided into two aliquots, which were incubated with an anti-LEF1 antibody (1:50, Cell Signaling #76010) and anti-rabbit IgG antibody (1:1000, or 1 μL for one immunoprecipitation, Cell Signaling #2729) as a negative control, respectively. The antibody-bound complex was precipitated with Dynabeads protein G (Invitrogen #10004D). The DNA fragments were released from the immunoprecipitated complexes by reversing the cross-linking at 65 °C overnight. The precipitated DNA was isolated, purified, and used as a template for PCR. The primers used for ChIP-qPCR were listed in Table 2.

### 4.9. Silencing Experiments

The siRNA plasmids consist of a pool of three human LEF1-targeting sequences (Santa Cruz, sc-35804), and the control scrambled siRNA (Santa Cruz, sc-37007) were purchased from Santa Cruz. Transient transfections were carried out using Lipofectamine RNAiMAX Transfection Reagent (Invitrogen, Waltham, MA, USA) following the manufacturer’s protocol. After 48 h of transfection, cells were collected for RT-qPCR and Western blotting analysis.

### 4.10. Statistical Analysis

Statistical analyses were performed using GraphPad Prism 7 software. Data were analyzed using unpaired Student’s *t*-test for two groups. One-way and two-way analysis of variance (ANOVA) followed by post hoc multiple comparisons were used to compare the means of three or more groups. The analysis method adapted in each experiment was indicated in figure legends. All experiments were repeated at least three times, and the results are presented as the mean ± SD. A *p* value < 0.05 or <0.1 was considered significant.

## Figures and Tables

**Figure 1 ijms-23-05499-f001:**
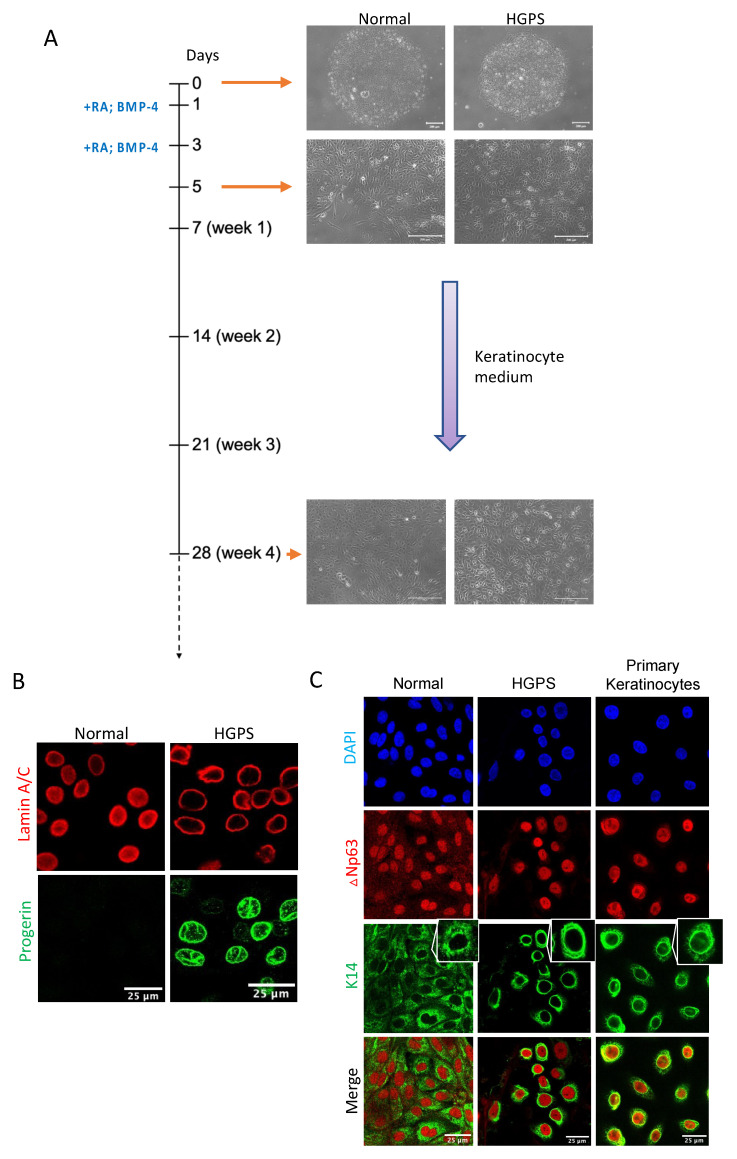
Differentiating normal and HGPS-derived iPSCs into keratinocytes. (**A**) Schematic representation of the protocol for differentiating normal and HGPS patient-derived iPSCs into a keratinocyte lineage. RA and BMP-4 were added on day 1 and day 3. Cells were maintained in keratinocyte growth medium for more than 28 days (4 weeks). Representative phase-contrast images of normal and HGPS iPSCs at different time points during keratinocyte induction were shown next to the timeline (scale bar = 200 μm). (**B**) Lamin A/C and progerin expression in normal and HGPS iPSCs differentiated cells at week 4 indicated by immunofluorescence staining (scale bar = 25 μm). (**C**) Basal layer keratinocyte markers ∆Np63 and K14 expression in normal and HGPS iPSCs differentiated cells at week 4 as well as in primary keratinocytes indicated by immunofluorescence staining. Representative keratin 14 filaments from single selected cells were shown in the upper-right corner (scale bar = 25 μm). The fluorescence intensities of protein markers were auto-adjusted to demonstrate their cellular distributions.

**Figure 2 ijms-23-05499-f002:**
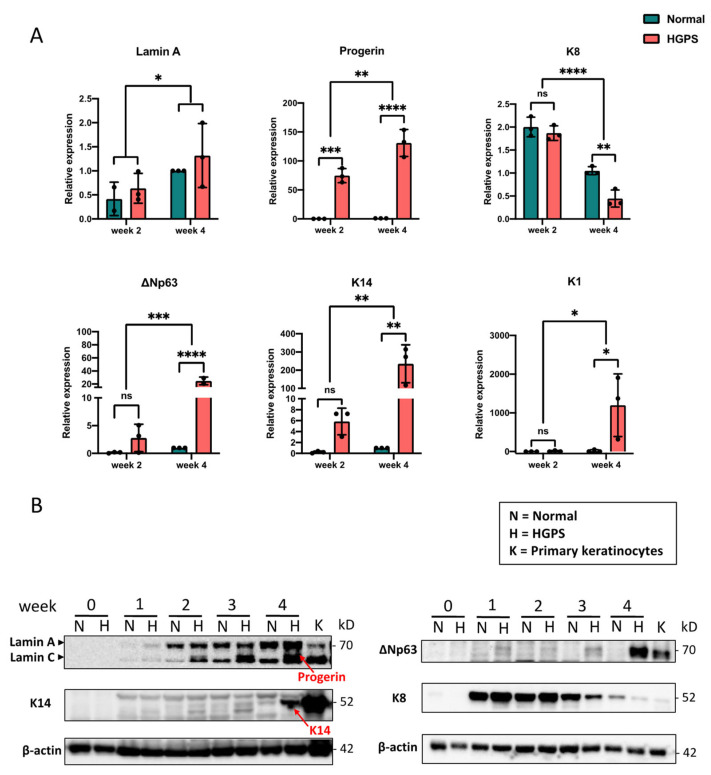
The expression of keratinocyte markers during iPSCs-keratinocytes differentiation. (**A**) Quantitative RT-PCR analysis of the relative expression of Lamin A, progerin, ∆Np63, K14, K8, and K1 during normal and HGPS iPSCs-keratinocytes induction. Data were normalized to endogenous *ACTB* mRNA and to the average of Normal week 4. Data are presented as mean ± SD (*n* = 3). * *p* < 0.05, ** *p* < 0.01, *** *p* < 0.001, **** *p* < 0.0001, ns, not significant, two-way ANOVA followed by Tukey’s multiple comparisons test. (**B**) Western blot analysis of Lamin A/C (including progerin in HGPS-differentiated cells indicated by the red arrow), ∆Np63, K14 (indicated by red arrow), and K8 expression during normal and HGPS iPSCs-keratinocytes induction. Primary keratinocytes were used as a positive control. All experiments were repeated at least three times, and representative data are shown as indicated.

**Figure 3 ijms-23-05499-f003:**
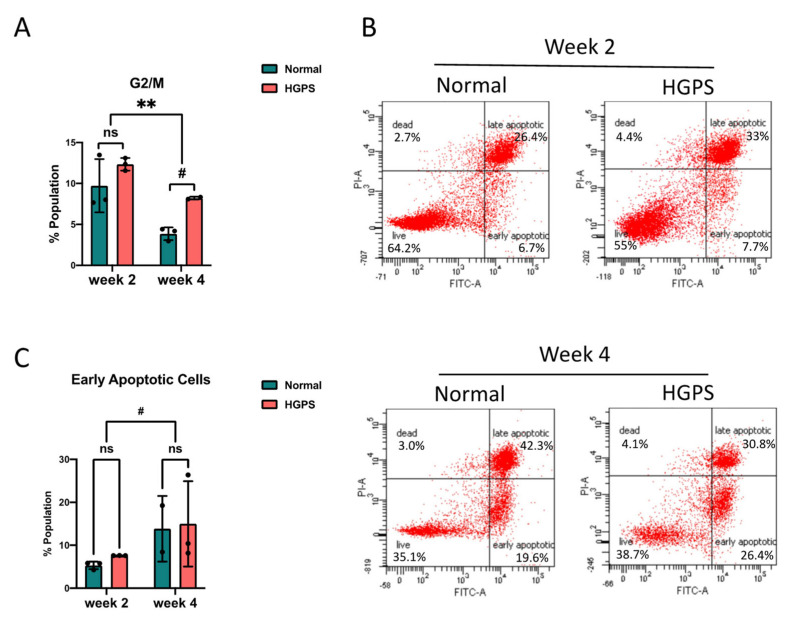
Cell cycle and apoptosis during iPSCs-keratinocytes differentiation (**A**) Quantification of the G2/M phase cell cycle partitioning during normal and HGPS iPSCs-keratinocytes induction. Data are presented as mean ± SD (*n* = 3). # *p* < 0.1, ** *p* < 0.01, ns, not significant, two-way ANOVA followed by Tukey’s multiple comparisons test. (**B**) Representative flow cytometry plots showing PI–annexin V apoptosis assay during normal and HGPS iPSCs-keratinocytes induction. The gates were set according to the positive and negative controls as per the manufacturer’s instructions. The cells in the lower right quadrant were quantified as early apoptotic populations. (**C**) Percentage of early apoptotic cells by PI–annexin V flow cytometry analysis during normal and HGPS iPSCs-keratinocytes induction. Data are presented as mean ± SD (*n* = 3). # *p* < 0.1, ns, not significant, Two-way ANOVA followed by Tukey’s multiple comparisons test.

**Figure 4 ijms-23-05499-f004:**
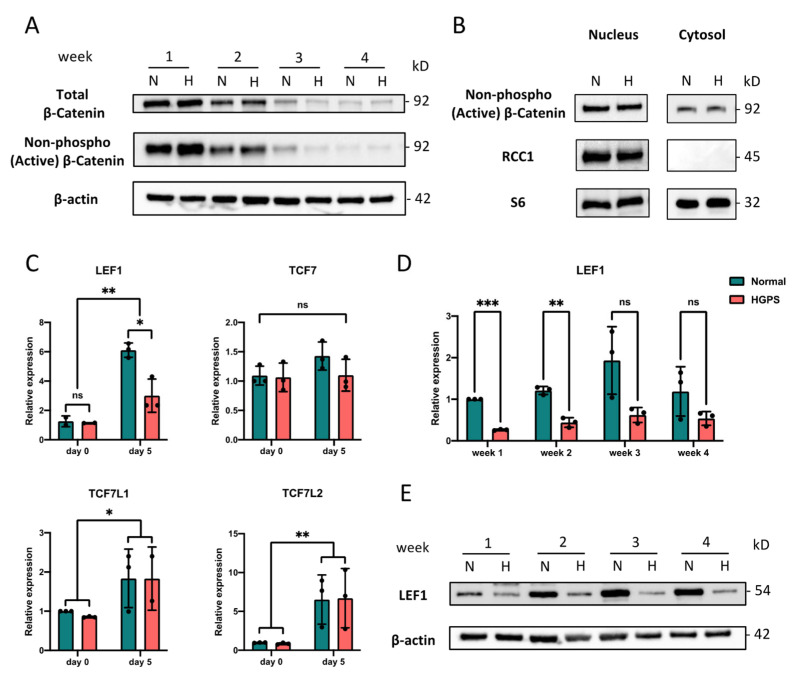
LEF1 down-regulation in early HGPS iPSCs-keratinocytes differentiation. (**A**) Western blot analysis of total and non-phospho (active) β-catenin expression during normal and HGPS iPSCs-keratinocytes induction. Experiments were repeated at least three times, and representative data are shown as indicated. (**B**) Western blot analysis of nuclear and cytosolic non-phospho (active) β-catenin level in early-differentiating (week 1) normal and HGPS iPSCs. Regulator of Chromosome Condensation 1 (RCC1) and S6 were used as a nuclear marker and the overall loading control, respectively. Experiments were repeated at least three times, and representative data are shown as indicated. (**C**) Quantitative RT-PCR analysis of the relative expression of WNT transcription factors LEF1, TCF7, TCF7L1, and TCF7L2 before (day 0) and after induction with RA and BMP-4 (day 5) in normal and HGPS iPSCs differentiation. Data were normalized to endogenous ACTB mRNA and to the average of Normal day 0. Data are presented as mean ± SD (*n* = 3). * *p* < 0.05, ** *p* < 0.01, ns, not significant, two-way ANOVA followed by Tukey’s multiple comparisons test. (**D**) Quantitative RT-PCR analysis of the relative expression of LEF1 during normal and HGPS iPSCs-keratinocytes induction. Data were normalized to endogenous ACTB mRNA and to the average of Normal week 1. Data are presented as mean ± SD (*n* = 3). ** *p* < 0.01, *** *p* < 0.001, ns, not significant, two-way ANOVA followed by Sidak’s multiple comparisons test. (**E**) Western blot analysis of LEF1 expression during normal and HGPS iPSCs-keratinocytes induction. Experiments were repeated at least three times, and representative data are shown as indicated.

**Figure 5 ijms-23-05499-f005:**
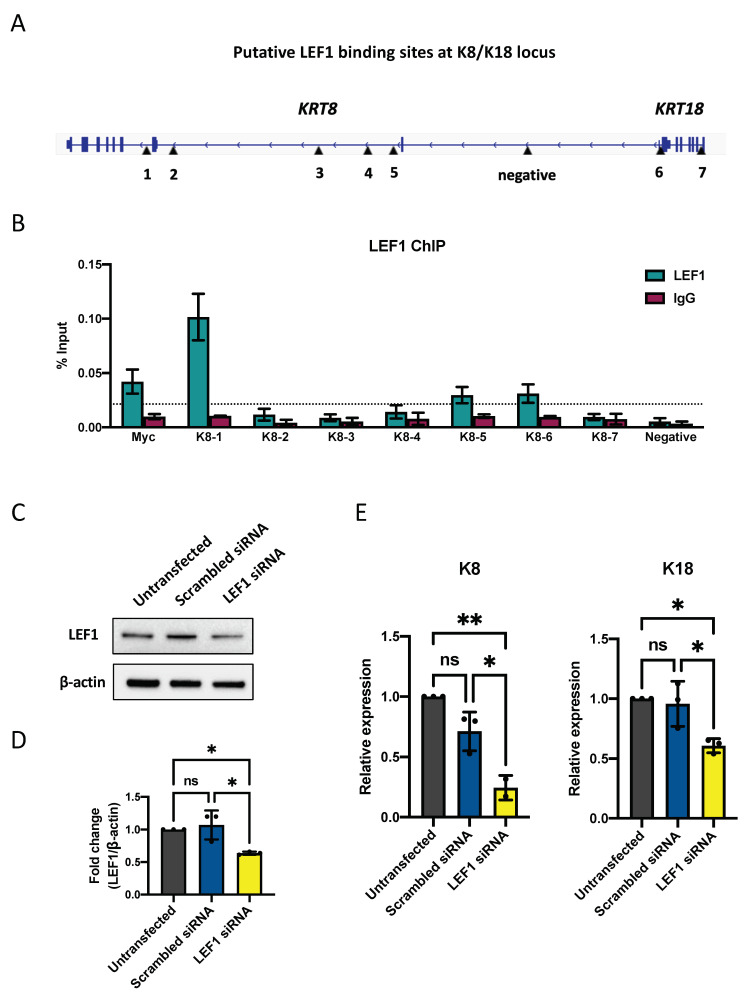
LEF1 regulates keratinocytes differentiation through K8/K18. (**A**) Schematic representation showing putative LEF1 binding sites at K8/K18 gene locus. The position of seven putative LEF1 binding sites and a non-specific locus (negative) were indicated by black arrowheads. (**B**) Chromatin immunoprecipitation quantitative PCR (ChIP-qPCR) analysis of the DNA binding activity of LEF1 at the K8/K18 locus. A known LEF1 binding site at the MYC gene was used as a positive control. The binding sites with LEF1 enrichment four-fold greater than the non-specific locus (above the dashed line) were considered to have strong LEF1 binding. Data are presented as mean ± SD (*n* = 3). (**C**) Representative Western blot result showing LEF1 protein expression 48 h after LEF1 siRNA knockdown in normal iPSCs differentiation. (**D**) Quantification of LEF1 siRNA knockdown Western blot analysis in (**C**). Data are presented as mean ± SD (*n* = 3). * *p* < 0.05, ns, not significant, one-way ANOVA followed by Tukey’s multiple comparisons test. (**E**) Quantitative RT-PCR analysis of the relative expression of K8 and K18 48 h after LEF1 siRNA knockdown in normal iPSCs differentiation. Data are normalized to endogenous ACTB mRNA and to the average of untransfected samples. Data are presented as mean ± SD (*n* = 3). * *p* < 0.05, ** *p* < 0.01, ns, not significant, one-way ANOVA followed by Tukey’s multiple comparisons test.

**Figure 6 ijms-23-05499-f006:**
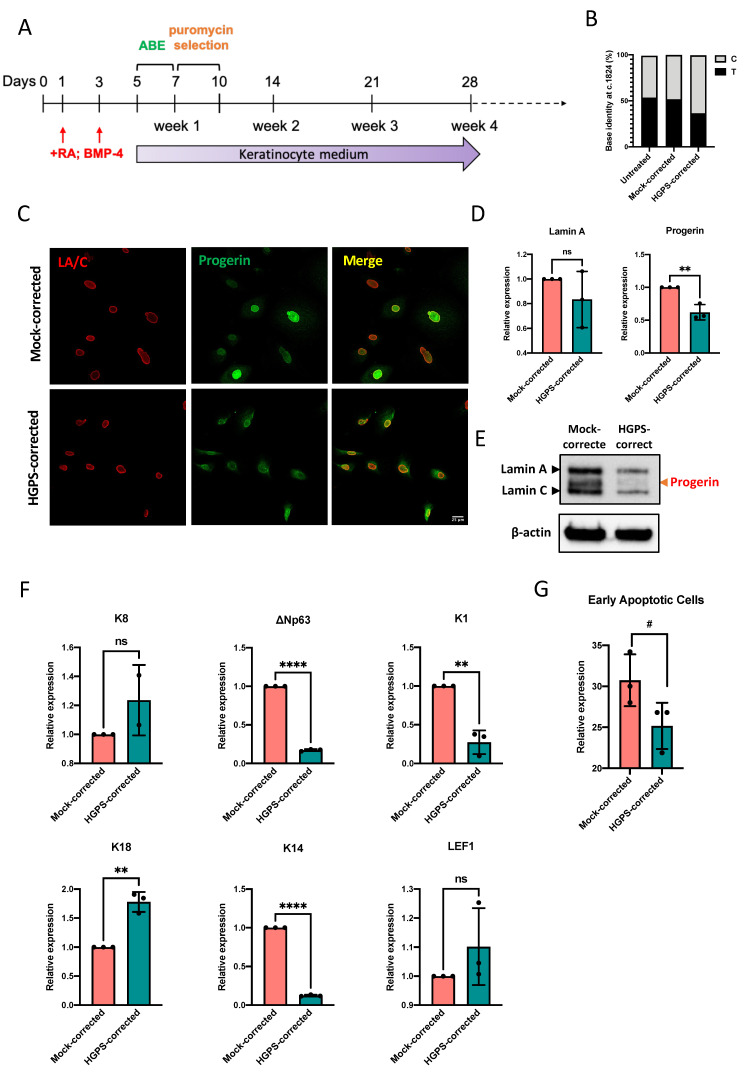
ABE corrects the HGPS mutation in iPSCs-derived keratinocytes. (**A**) Schematic representation of the Adenine base editing in HGPS patient-derived iPSCs-keratinocytes induction. (**B**) LMNA c.1824 nucleotide identity in HGPS iPSCs derived-keratinocytes untreated or treated with ABE7.10max-VRQR lentivirus (mock and HGPS mutation-targeting) at differentiation week 4. (**C**) Lamin A/C and progerin expression in mock-corrected and HGPS-corrected iPSCs-derived keratinocytes after week 4 (~day 35, passage 1) indicated by immunofluorescence staining (scale bar = 25 μm). (**D**) Quantitative RT-PCR analysis of the relative expression of Lamin A and progerin in mock-corrected and HGPS-corrected iPSCs-derived keratinocytes at week 4. Data were normalized to endogenous ACTB mRNA and to the average of mock-corrected cells. Data are presented as mean ± SD (*n* = 3). ** *p* < 0.01, ns, not significant, unpaired two-tailed *t*-test. (**E**) Western blot analysis of Lamin A/C and progerin expression in mock-corrected and HGPS-corrected iPSCs-derived keratinocytes at week 4. Experiments were repeated at least three times, and representative data are shown as indicated. (**F**) Quantitative RT-PCR analysis of the relative expression of ∆Np63, K14, K8, K18, K1, and LEF1 in mock-corrected and HGPS-corrected iPSCs-derived keratinocytes at week 4. Data were normalized to endogenous ACTB mRNA and to the average of mock-corrected cells. Data are presented as mean ± SD (*n* = 3). ** *p* < 0.01, **** *p* < 0.0001, ns, not significant, unpaired two-tailed *t*-test. (**G**) Percentage of early apoptotic cells by PI–annexin V flow cytometry analysis in mock-corrected and HGPS-corrected iPSCs-derived keratinocytes at week 4. Data are presented as mean ± SD (*n* = 3). # *p* < 0.1, unpaired two-tailed *t*-test.

**Figure 7 ijms-23-05499-f007:**
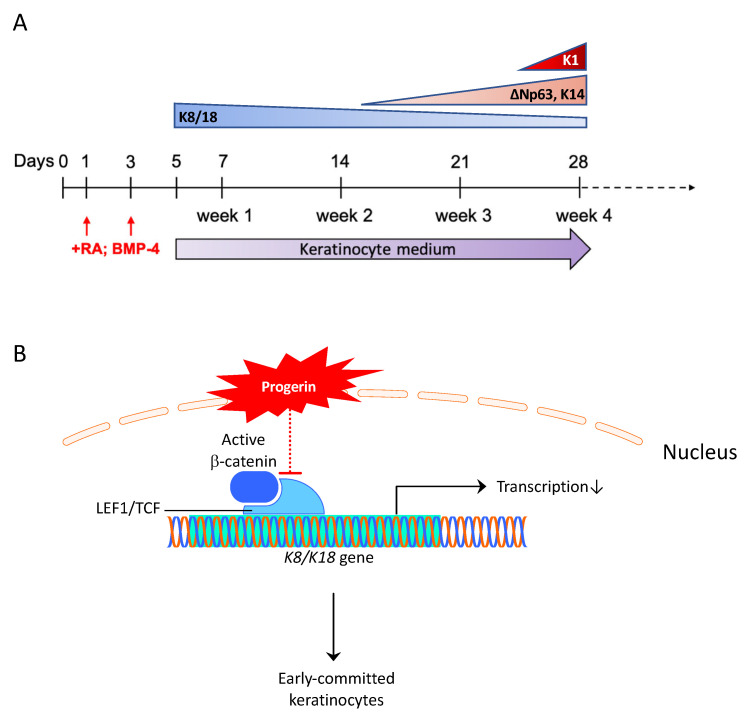
Working model of epidermal development in HGPS. (**A**) Schematic representation of the keratinocyte-related markers succession during iPSCs-keratinocytes induction. (**B**) Schematic diagram showing the molecular mechanism for the early commitment to the keratinocyte lineage in HGPS iPSCs differentiation.

**Table 1 ijms-23-05499-t001:** Primer sequences used for RT-qPCR experiments.

Gene		Sequence
ACTB	Forward	5′-CTGGAACGGTGAAGGTGACA-3′
	Reverse	5′-AAGGGACTTCCTGTAACAATGCA-3′
LMNA	Forward	5′-GCAACAAGTCCAATGAGGACCA-3′
	Reverse	5′-CATGATGCTGCAGTTCTGGGGGCTCTGGAT-3′
Progerin	Forward	5′-GCAACAAGTCCAATGAGGACCA-3′
	Reverse	5′-CATGATGCTGCAGTTCTGGGGGCTCTGGAC-3′
K8	Forward	5′-CAGAAGTCCTACAAGGTGTCCA-3′
	Reverse	5′-CTCTGGTTGACCGTAACTGCG-3′
K18	Forward	5’-GGCATCCAGAACGAGAAGGAG-3’
	Reverse	5’-ATTGTCCACAGTATTTGCGAAGA-3’
∆Np63	Forward	5’-ACCTGGAAAACAATGCCCAGA-3’
	Reverse	5’-GGCAATCTGTCCCTCGTTGA-3’
K14	Forward	5’-TGAGCCGCATTCTGAACGAG-3’
	Reverse	5’-GATGACTGCGATCCAGAGGA-3’
K1	Forward	5′-GGTGCTTATATGACCAAGGTGG-3′
	Reverse	5′-ATGCTGTCCAGGTCGAGACT-3′
LEF1	Forward	5′-CGATGACGGAAAGCATCCAG-3′
	Reverse	5′-CCACCCGGAGACAAGGGATA-3′
TCF7	Forward	5′-AACATTTCAACAGCCCACATCC-3′
	Reverse	5′-GGAGTAGAAGCCAGAGAGGTC-3′
TCF7L1	Forward	5′-CCACAGTCAAGGACACGAGG-3′
	Reverse	5′-TCGATCTCTGGGGAGAGGTG-3′
TCF7L2	Forward	5′-CCTCACGCCTCTTATCACGTA-3′
	Reverse	5′-AGGCGATAGTGGGTAATACGG-3′

**Table 2 ijms-23-05499-t002:** Primer sequences used for ChIP-qPCR experiments.

LEF1 Binding Site		Sequence
MYC	Forward	5′-CCCAAAAAAAGGCACGGAA-3′
	Reverse	5′-TATTGGAAATGCGGTCATGC-3′
K8-1	Forward	5′-TCCCAGGAGCCAGTCAGCGT-3′
	Reverse	5′-GCCCATTCAGCCCGTGTCCC-3′
K8-2	Forward	5′-GCTCCCAACGGGCCAGAGGA-3′
	Reverse	5′-GCCTGGGGTGCCTTGCTCAG-3′
K8-3	Forward	5′-ACCTTCAGACCACCCTCCCCC-3′
	Reverse	5′-ACCCTGGGAAGAGGCAGCAGA-3′
K8-4	Forward	5′-CGTCCTGTGAACTGACCCTCCC-3′
	Reverse	5′-TGTTTTCCTGCTACTGCACA-3′
K8-5	Forward	5′-CCCCTCCGTCATCCTGGCCT-3′
	Reverse	5′-CCTGGGGCTGGCTGGTAGGA-3′
K8-6	Forward	5′-CTGCGCCCGGTCAAGACTCC-3′
	Reverse	5′-CCGGGCATGGACACGGACAG-3′
K8-7	Forward	5′-CACAGACCCGGGCAGAGGGA-3′
	Reverse	5′-TCTTCCAGCAGGCGGCGGTA-3′
Negative	Forward	5′-GCCACTTGGGAGGCTGAGGC-3′
	Reverse	5′-TTGAGCCACTGCACCCTGCC-3′

## Data Availability

Not applicable.

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
