# Peer review of "Impaired LEF1 Activation Accelerates iPSC-Derived Keratinocytes Differentiation in Hutchinson-Gilford Progeria Syndrome"

_ijms, 2022, doi:10.3390/ijms23105499_

Round 1
Reviewer 1 Report
The authors report progerin-induced alterations in keratinocyte differentiation using human HGPS iPSC as a model system. Although previous in vivo studies have shown that HGPS mice show keratinocyte alterations, the present work adds mechanistic insight into the process using cells derived from HGPS patients, which adds relevance to the study.
Major comments:
- The causal role of reduced LEF1 levels in the downregulation of K8 in HGPS cells is not clear. Although Fig. 5E shows that LEF1 knockdown in control cells reduces K8 expression, Fig. 1A shows no difference in K8 expression between control and HGPS cells at 2 weeks, a time point when HGPS cells already show reduced LEF1 levels (Fig. 4D). These data suggest that LEF1 reduction is not sufficient to reduce K8 levels in HGPS cells. Maybe a more direct and complementary approach could be to overexpress LEF1 in HGPS cells at 4 weeks, and assess whether K8 levels are recovered.
- The authors imply that K8/K18 reduction in HGPS cells may have a causal role in the differentiation alterations observed in HGPS cells (lines 336-338 and Fig. 7). Is that the case? Assessment of K1 expression after K8/K18 knockdown in control cells could help to address this point. Otherwise, the discussion should be rephrased.
Minor comments:
- Lines 83 and 358: in vivo ABE editing (ref. 16, Koblan et al) was not carried out during embryonic development, but postnatally (P3 and P14). Please correct.
- Fig. 1C: nuclear staining should be included to assess the percentage of cells expressing basal layer keratinocyte makers (especially in the images showing HGPS cells, where it is not obvious that all the field is filled with cells). In addition, this panel does not recapitulate the differences observed in ∆Np63 and K14 levels between control and HGPS cells shown by western blot (4 week time point) in figure 2B. Could you please explain the discrepancy?
- Lines 358-363: please do not confuse base editing efficiency with the efficiency of progerin clearance, since they are two different things. Please rephrase.
- The authors should mention in the introduction the lack of phenotype (except nuclear abnormalities) in the skin of mice expressing progerin under the K14 promoter (ref. 11, Wang et al), putting it into the context of the other mouse models that do develop a phenotype.
- Fig. 5 and 6. Some asterisks showing statistical significance have been erased. Please correct.
- Minor English editing is needed throughout the abstract and main text.
Author Response
Reviewer 1
Major comments:
- The causal role of reduced LEF1 levels in the downregulation of K8 in HGPS cells is not clear. Although Fig. 5E shows that LEF1 knockdown in control cells reduces K8 expression, Fig. 1A shows no difference in K8 expression between control and HGPS cells at 2 weeks, a time point when HGPS cells already show reduced LEF1 levels (Fig. 4D). These data suggest that LEF1 reduction is not sufficient to reduce K8 levels in HGPS cells. Maybe a more direct and complementary approach could be to overexpress LEF1 in HGPS cells at 4 weeks, and assess whether K8 levels are recovered.
We thank the reviewer for pointing out that the LEF1 reduction at week 2 has not shown its impact on K8 expression in HGPS cells yet. We think one possible explanation is that it may take a longer time for the K8 protein down-regulation to become readily detectable, as LEF1 is still induced in HGPS iPSC differentiation, but to a lesser extent in comparison to control cells.
Overexpressing LEF1 in HGPS cells will be a great approach to test its impact on K8 expression. Due to the strict revision time of this manuscript, we do not have sufficient time to construct an overexpression plasmid of LEF1, conduct differentiation experiments, and analyze K8 expression. We have added this to the discussion as an important future experiment (highlighted).
- The authors imply that K8/K18 reduction in HGPS cells may have a causal role in the differentiation alterations observed in HGPS cells (lines 336-338 and Fig. 7). Is that the case? Assessment of K1 expression after K8/K18 knockdown in control cells could help to address this point. Otherwise, the discussion should be rephrased.
In our differentiation experiment, the K1 upregulation in HGPS iPSCs-keratinocytes was detected one week after the down-regulation of K8, but absent in the normal control samples (Figure 2A &B and Figure S1). We have rephrased our discussion in section 3.2.
Minor comments:
- Lines 83 and 358: in vivo ABE editing (ref. 16, Koblan et al) was not carried out during embryonic development, but postnatally (P3 and P14). Please correct.
Thank you. We made the corrections in the manuscript (highlighted).
- Fig. 1C: nuclear staining should be included to assess the percentage of cells expressing basal layer keratinocyte makers (especially in the images showing HGPS cells, where it is not obvious that all the field is filled with cells). In addition, this panel does not recapitulate the differences observed in ∆Np63 and K14 levels between control and HGPS cells shown by western blot (4 week time point) in figure 2B. Could you please explain the discrepancy?
Thank you. We added the DAPI channel of nuclear staining in the updated Figure 1C. The fluorescence intensities of protein markers shown in Figure 1 are not quantitative, which were auto-adjusted by the imageJ software to best illustrate the signals. We added the signal intensity statement in the figure legend (Highlighted).
- Lines 358-363: please do not confuse base editing efficiency with the efficiency of progerin clearance, since they are two different things. Please rephrase.
Thank you for this important point. We have rephrased the manuscript (highlighted in section 3.3).
- The authors should mention in the introduction the lack of phenotype (except nuclear abnormalities) in the skin of mice expressing progerin under the K14 promoter (ref. 11, Wang et al), putting it into the context of the other mouse models that do develop a phenotype.
Thank you. We included this study in the introduction (highlighted).
- Fig. 5 and 6. Some asterisks showing statistical significance have been erased. Please correct.
In Figure 5B, instead of performing a student t-test, we defined strong LEF1 binding as four-fold greater than the enrichment of the non-specific locus (above the dashed line), which was suggested by the Cell Signaling ChIP protocol. Thus, we did not label the asterisks in this figure and explained the rubrics in the figure legend (highlighted).
In Figure 6B, the percentage of base identity at LMNA c.1824 was acquired from one sequencing result, thus, no statistics were performed.
Reviewer 2 Report
In the submitted manuscript Mao and colleagues use differentiation of human patient derived iPSC to keratinocyte lineage to study early skin development that could contribute to severe skin phenotypes in HGPS. Authors show accelerated keratinocyte lineage commitment in HGPS, implicate increased apoptosis in these cells and focus on Wnt signalling as an underlying molecular cause. Mechanistically, authors show reduced levels of downstream transcription factor LEF1 without alterations of upstream beta-catenin total levels nor changes in its localization and active state. Nevertheless, authors show clearly gene regulatory function of LEF1 in activation of K18/8 genes and partial rescue of the differentiation defects using ABE editing to reduce progerin levels. Even though there are some reports on defective Wnt Signalling in HGPS this manuscript fills the missing gap regarding Wnt signalling during early skin development which is highly relevant for the field.
The concept and experimental design is in general appropriate. Major concern is however that on the mechanistic level ; LEF1 diminished activity as an underlying cause for the defects is not fully clear. If LEF1 is causative for the K8/18 effect why were the levels not rescued using ABE editing? Perhaps LEF1 at earlier time points needs to be tested? Furthermore, if no changes in beta-catenin levels observed, what causes then the observed phenotype? Do authors observe changes in other upstream components of Wnt pathway such as Wnt or GSK previously shown to be affected in HGPS (Hernandez et al., 2010, Dev Cell)? Also, ChIP experiments described in Figure 5 need to be performed in HGPS cells in order to assess K8 binding sites and obtain a clearer picture.
Specific major comments
- Progerin bands in Figure 2B are hardly visible particularly at earlier time points. Authors should provide images with higher resolution or exposure similarly to Figure 6E. Alternatively immunofluorescence images at earlier time points using progerin antibodies could provide more explanation. This is important in view of the effect that at earlier time points <2weeks progerin needs to be expressed in order to cause the effects. Furthermore, why does progerin show unusual dotty staining in Figure 1B in contrast to more usual mock-corrected control in Figure 6C?
- Similarly, many different “unspecific” K14 protein bands are shown in Figure 2B . Authors should indicate at least specific K14 bands with an arrow.
- The authors statements in Figure 3 regarding higher percentage of dying cells in HGPS cells (lane 181) and statement of massive cell death in HGPS (Figure 6G, lane 283) seem somehow conflicting. First, at week 2 time point appears to be no significant differences in the percentages of early and late apoptotic cells and similarly at week 4, where trends regarding the late apoptotic states appear to go even in the opposite direction, higher number of apoptotic cells in “normal “ versus HGPS. Additional data or explanations for these statements are necessary. Authors should also at least comment why is the number of viable cells so low in both normal and HGPS samples ( ~35%) at week 4 in Figure 4C and ditto why is the apoptotic rate so high in mock-corrected samples in Figure 6G (compared to Figure 3C)? Did authors check senescence markers e.g. p16 and inflammation?
Minor comments
-Representative images should show cells at similar confluence in all three panels otherwise comparisons are not justified. Also high power images of keratin network in single selected cells that could be shown in the corner would be preferential. Details on cell culture should be included. Do authors passage cells during the 28 day differentiation and/or show cells in any of the experiments after passaging?
-For all Western blots the indication of protein the size of molecular weight marker bands should be indicated.
-Source and isolation of primary keratinocytes in Figure 2B used as controls should be described in Method section.
-Abbreviation for RCC1 used as nuclear loading control in Figure 4B should be stated in the legend.
-p values in diagrams should be indicated by stars Figure 5 D-E.
-In the introduction and discussion previous data on Wnt signalling in HGPS models should be included and discussed Ref. Hernandez et al., 2010, Dev Cell.
- “normal “ is inappropriate characterization please replace in the whole text with “unaffected” or “healthy “ father donor
-References for statements in following lanes should be provided otherwise statements should be modified:
- Lane 311 EPC depletion for HGPS; Lanes 328-329 Reference and which WNT pathway components are induced upon iPSCs-keratinocytes (RA and BMP) differentiation should be included.
Author Response
Reviewer 2
Comments and Suggestions for Authors
In the submitted manuscript Mao and colleagues use differentiation of human patient derived iPSC to keratinocyte lineage to study early skin development that could contribute to severe skin phenotypes in HGPS. Authors show accelerated keratinocyte lineage commitment in HGPS, implicate increased apoptosis in these cells and focus on Wnt signalling as an underlying molecular cause. Mechanistically, authors show reduced levels of downstream transcription factor LEF1 without alterations of upstream beta-catenin total levels nor changes in its localization and active state. Nevertheless, authors show clearly gene regulatory function of LEF1 in activation of K18/8 genes and partial rescue of the differentiation defects using ABE editing to reduce progerin levels. Even though there are some reports on defective Wnt Signalling in HGPS this manuscript fills the missing gap regarding Wnt signalling during early skin development which is highly relevant for the field.
The concept and experimental design is in general appropriate. Major concern is however that on the mechanistic level ; LEF1 diminished activity as an underlying cause for the defects is not fully clear. If LEF1 is causative for the K8/18 effect why were the levels not rescued using ABE editing? Perhaps LEF1 at earlier time points needs to be tested?
We appreciate these insightful comments. It is a great suggestion to detect LEF1 expression at earlier time points. However, in this study, we conducted ABE correction during iPSCs-keratinocytes differentiation (Figure 6A), and it takes a few cell cycles for the ABE mechanism to correct progerin mutation in the system, thus we could not check LEF1 at the earlier time point. Looking back, a possible better approach is to correct the HGPS mutation with ABE directly in iPS cells. We have added this to the end of the discussion (highlighted).
Furthermore, if no changes in beta-catenin levels observed, what causes then the observed phenotype? Do authors observe changes in other upstream components of Wnt pathway such as Wnt or GSK previously shown to be affected in HGPS (Hernandez et al., 2010, Dev Cell)? Also, ChIP experiments described in Figure 5 need to be performed in HGPS cells in order to assess K8 binding sites and obtain a clearer picture.
Although it is unclear how LEF1 transcription was inhibited at this stage of differentiation, other potential pathways to regulate LEF1 expression in a β-catenin-independent manner are present, in particular, the BMP pathway that activates LEF1 transcription via P-SMAD in the neural stem cells differentiation. We also haven’t examined upstream components, such as Wnt or GSK, in our iPSCs-keratinocytes differentiation. This is a great suggestion and we will add it to future experiments.
As for the suggested ChIP experiment, due to the decreased expression of LEF1 in HGPS cells, we were unable to use the LEF1 antibody to IP a sufficient amount of genomic DNA for quantitative PCR analysis.
Specific major comments
Progerin bands in Figure 2B are hardly visible particularly at earlier time points. Authors should provide images with higher resolution or exposure similarly to Figure 6E. Alternatively immunofluorescence images at earlier time points using progerin antibodies could provide more explanation. This is important in view of the effect that at earlier time points <2weeks progerin needs to be expressed in order to cause the effects. Furthermore, why does progerin show unusual dotty staining in Figure 1B in contrast to more usual mock-corrected control in Figure 6C?
We have replaced Figure 2B with a longer exposure image. We also included a Western blot with progerin antibody in Figure S1A showing progerin expression at earlier time points. Furthermore, we have replaced Figure 1B as suggested for a better representative image.
Similarly, many different “unspecific” K14 protein bands are shown in Figure 2B . Authors should indicate at least specific K14 bands with an arrow.
Thank you. We now indicated the specific K14 band with an arrow in updated Figure 2B.
The authors statements in Figure 3 regarding higher percentage of dying cells in HGPS cells (lane 181) and statement of massive cell death in HGPS (Figure 6G, lane 283) seem somehow conflicting. First, at week 2 time point appears to be no significant differences in the percentages of early and late apoptotic cells and similarly at week 4, where trends regarding the late apoptotic states appear to go even in the opposite direction, higher number of apoptotic cells in “normal “ versus HGPS. Additional data or explanations for these statements are necessary. Authors should also at least comment why is the number of viable cells so low in both normal and HGPS samples ( ~35%) at week 4 in Figure 4C and ditto why is the apoptotic rate so high in mock-corrected samples in Figure 6G (compared to Figure 3C)? Did authors check senescence markers e.g. p16 and inflammation?
Thank you! Instead of late apoptotic cells, we prefer to use early apoptotic cells as an indication of cell death because it quantifies the cells with phosphotidylserine externalization and will be less affected by perturbations from experimental procedures (such as detaching the cells from plating) that damage the cell membrane. In our three independent biological replicates, the variations for early apoptotic cells at week 4 are huge. However, in each individual experiment, the percentage of early apoptotic cells was always higher in HGPS than in normal. Thus, we concluded that higher percentage of dying cells in HGPS samples. We have made this statement clearer in the revision.
High apoptosis and cell turn-over rate are characteristics of normal skin tissue. In our in vitro iPSCs differentiation system, the cell harvesting procedure includes multiple spins that may inevitably damage the cell membrane and increase the population of late apoptotic cells. In addition, viral transduction and puromycin selection in ABE experiment may also affect cell viability and increase cell death rate.
We have checked the senescence marker p16, but did not observe a significant difference during the iPSCs-keratinocytes differentiation (data not shown).
Minor comments
-Representative images should show cells at similar confluence in all three panels otherwise comparisons are not justified. Also high power images of keratin network in single selected cells that could be shown in the corner would be preferential. Details on cell culture should be included. Do authors passage cells during the 28 day differentiation and/or show cells in any of the experiments after passaging?
Thank you. We have addressed all these concerns in updated Figure 1. We noticed that normal iPSCs-derived keratinocytes are more confluent than HGPS at the end of differentiation, thus the cell density of normal iPSCs-keratinocytes is relatively higher. The cells were not passaged during the 28-day differentiation. They can be passaged after four weeks. Figure 6C shows the passaged iPSCs-keratinocytes. We added more details on cell culture in Figure 6 legend and the method section 4.1 (highlighted).
-For all Western blots the indication of protein the size of molecular weight marker bands should be indicated.
Thank you! We added the molecular weight of marker bands in updated Figures 4-6.
-Source and isolation of primary keratinocytes in Figure 2B used as controls should be described in Method section.
We added the primary keratinocytes information in Methods section 4.1 (highlighted).
-Abbreviation for RCC1 used as nuclear loading control in Figure 4B should be stated in the legend.
We added the abbreviation for RCC1 in Figure 4 legend (highlighted).
-In the introduction and discussion previous data on Wnt signalling in HGPS models should be included and discussed Ref. Hernandez et al., 2010, Dev Cell.
We added this reference in both introduction and discussion (highlighted).
- “normal “ is inappropriate characterization please replace in the whole text with “unaffected” or “healthy “ father donor
We included a statement indicating “normal” means cells from healthy/unaffected father donor in line 90 and Methods section 4.1 (highlighted). We would like to keep the term “normal” in the manuscript in order to be consistent with previous publications.
-References for statements in following lanes should be provided otherwise statements should be modified:
- Lane 311 EPC depletion for HGPS; Lanes 328-329 Reference and which WNT pathway components are induced upon iPSCs-keratinocytes (RA and BMP) differentiation should be included.
References have been updated according to the suggestion.
Round 2
Reviewer 1 Report
The authors did not carry out any of the additional experiments proposed. Although I do not agree with this kind of review, I understand that it is consequence of the short time given by the Journal to address the issues raised by reviewers. On the other hand, the authors have adequately rephrased some sections and completed missing information.
Minor comments:
- Line 69: replace progeroic by progeroid.
- Fig. 5E, 5D, 6D, 6F: non significant results are labeled as ns, but it looks like significance asterisks (mentioned in the figure legends) are still missing, probably a typo. Please correct.
Author Response
- Line 69: replace progeroic by progeroid.
Thank you. We have made the correction (Highlighted).
- Fig. 5E, 5D, 6D, 6F: non significant results are labeled as ns, but it looks like significance asterisks (mentioned in the figure legends) are still missing, probably a typo. Please correct.
We can see the asterisks on the referred figures. Could the reviewer download the manuscript to see if the asterisks are still missing?
Reviewer 2 Report
Authors have provided additional information and added higher exposures/magnification of images that provide for the most part clarification. However, authors statement regarding apoptosis in HGPS cells is still unclear (Figure 3C). There is no significant difference in apoptosis to corroborate authors statements and this leads to falsely conclusions. Authors should either omit the statement regarding the increased apoptosis in HGPS or include additional assays to assess this such as TUNEL assay or similar. Also senescence marker p16 data in normal and HGPS cells should be shown and included at least in the supplementary section.
Minor comment
Authors should show/mention in the text that LEF1 ChIP experiment in HGPS cells was performed but insufficient amounts of genomic DNA were obtained and add possible explanations.
Author Response
Comments and Suggestions for Authors
Authors have provided additional information and added higher exposures/magnification of images that provide for the most part clarification. However, authors statement regarding apoptosis in HGPS cells is still unclear (Figure 3C). There is no significant difference in apoptosis to corroborate authors statements and this leads to falsely conclusions. Authors should either omit the statement regarding the increased apoptosis in HGPS or include additional assays to assess this such as TUNEL assay or similar. Also senescence marker p16 data in normal and HGPS cells should be shown and included at least in the supplementary section.
Thank you. We have revised the apoptosis discussion in both the experimental result 2.2 and in discussion 3.1 (highlighted). The p16 Western blot data has been included in Figure S1A.
Minor comment
Authors should show/mention in the text that LEF1 ChIP experiment in HGPS cells was performed but insufficient amounts of genomic DNA were obtained and add possible explanations.
Please see revised section 2.4 (Highlighted).